# Hysteretic Behavior of Steel Reinforced Concrete Columns Based on Damage Analysis

**Bin Wang [1,*], Guang Huo [1], Yongfeng Sun [1] and Shansuo Zheng [2]**

[1] School of Civil and Architecture Engineering, Xi'an Technological University, Xi'an 710021, China; huoguang@163.com (G.H.); sunyongfeng@163.com (Y.S.)

[2] School of Civil Engineering, Xi'an University of Architecture and Technology, Xi'an 710055, China; zhengshansuo@263.net

[*] Correspondence: wangbin2346@xatu.edu.cn; Tel.: +86-135-1912-1853

**Abstract:** With the aim to model the seismic behavior of steel reinforced concrete (SRC) frame columns, in this research, hysteresis and skeleton curves were obtained based on the damage test results of SRC frame columns under low cyclic repeat loading and the hysteretic behavior of the frame columns was further analyzed. Then, the skeleton curve and hysteresis loops were further simplified. The simplified skeleton curve model was obtained through the corresponding feature points obtained by mechanical and regression analysis. The nonlinear combination seismic damage index, which was developed by the test results and can well reflect the effect of the loading path and the number of loading cycle of SRC frame columns, was used to establish the cyclic degradation index. The strength and stiffness degradation rule of the SRC frame columns was analyzed further by considering the effect of the accumulated damage caused by an earthquake. Finally, the hysteresis model of the SRC frame columns was established, and the specific hysteresis rules were given. The validity of the developed hysteresis model was verified by e comparison between the calculated results and the test results. The results showed that the model could describe the hysteresis characteristics of the SRC frame columns under cyclic loading and provide guidance for the elastoplastic time-history analysis of these structures.

**Keywords:** steel reinforced concrete; frame columns; hysteretic behavior; damage index; hysteresis model

## 1. Introduction

Due to their high bearing capacity and good seismic performance such as energy dissipation and ductility, steel reinforcement concrete (SRC) members, especially frame columns, are widely used in high-rise and super high-rise hybrid structures [1–4]. To ensure that the seismic performance of the structure meets the design requirements, the engineering designer must apply a strict elastoplastic calculation analysis of the structures. As the basis of elastoplastic calculation analysis, the hysteresis model that can accurately reflect the strength and stiffness degradation, deformation performance, and energy dissipation capacity of structures and members under cyclic repeated loading has been studied [5,6]. Generally, the existing hysteresis model includes two parts, namely, the skeleton curve and the hysteresis rule. The former is determined by all the hysteresis feature points and the latter reflects the highly nonlinear character of the structures [7]. For the structural seismic analysis, the existing hysteresis model can be divided into two categories: the polyline hysteresis (PH) model and the smooth hysteresis (SH) model. The PH model consists of piecewise lines obtained by simplifying the hysteresis behavior of the structure members, and is relatively easy to express and apply [8–11]. The SH model needs more computational burden to obtain more accurate results and is consequently

relatively difficult to implement [12,13]. Therefore, in this paper, the polyline hysteresis model was adopted for seismic analysis of SRC members.

Meanwhile, in most of the existing PH models, the strength and stiffness degradation rule of structure members under an earthquake is described by inducing the cyclic degradation index that only considers energy dissipation [14–17]. However, an earthquake is a random load, and the loading path has a significant effect on the ability of seismic energy consumption of the members. Therefore, it is difficult to obtain a uniform expression of the energy dissipation capacity of the members. Compared to methods mentioned above for establishing the hysteresis model, the hysteretic model of structural members is established by introducing a damage index (DI) that can better describe the stiffness and strength degradation rule [18,19].

At present, there are several damage analysis models for structure members. There are two main approaches to establish the damage index: one is sensor based and the other one is hysteresis based. The sensor-based DI often employs distributed sensors that are embedded or surface bonded to the structure to capture the change of certain physical parameters, which can reflect the severity of damages to the structure [20–24]. A commonly employed sensor is the Lead Zirconate Titanate (PZT) transducer in structural damage detection. PZT is a type of piezoceramic material with very strong piezoelectric effect that enables the PZT transducer to have dual actuation and sensing capacity. In addition, PZT transducers have wide bandwidth and high sensitivity, and are widely used to generate and detect stress waves [25–28]. With the active sensing approach, a pair of PZT sensors and an actuator located along a stress wave path can monitor the damage on the path and a damage index can be established to reflect the severity of the damage. The active-sensing based DI has been used in concrete structures [29–31], SRC structures [32–34], and concrete filled tubular structures [35–37]. Damage indices based on impedance have also been reported [38–40]. The advantage of the sensor-based DI is that it is suitable for real time monitoring of the structure via sensors integrated with the structure [41–43], however, this index does not directly relate to an inherent structural parameter.

The other approach to establish a damage index is based on hysteresis. The hysteretic curves of structure members reflect the changes of strength, stiffness, energy dissipation capacity, and displacement ductility with the loading displacement and the number of cycles. Therefore, the rule of strength and stiffness degradation of the members can be described by the change of the hysteresis loop under an earthquake. Early on, researchers proposed non-cumulative damage indexes based on a single parameter, such as stiffness, strength, and hysteretic energy dissipation [44–47]. Since an earthquake is a repeated loading process, the damage caused by an earthquake is cumulative. To better describe the damage of structures under cycle load, the cumulative seismic damage indexes, which consist of the combination of plastic deformation and cumulative hysteretic energy dissipation, were developed and are widely used to analyze structural damage [48–50]. Though the hysteresis-based DI involves a destructive test and cannot be used for real time monitoring of in-service structure, this damage index is directly link to inherent structural parameters, such as stiffness. In this research, we use the hysteresis-based DI to analyze the hysteresis rule of SRC columns.

The existing research results show that the cyclic loading history and the number of cycles are the main influencing factors for the damage analysis of structure members. However, the damage index selected in the existing hysteresis model cannot well consider the influence of the number of loading cycles and loading path on the ultimate energy dissipation and deformation capacity of the members. Meanwhile, to the best of knowledge of the authors, this method has been applied only to the analysis of RC structures and steel structures. Few researches on the hysteretic behavior of SRC members have been reported. Since the SRC composite members consist of steel and RC, their damage process is more complex than pure steel or reinforced concrete structures [51,52]. Therefore, the current method of establishing hysteresis model based DI cannot accurately reflect the seismic performance of the SRC members.

In this paper, a new PH hysteretic model by considering the accumulated damage is proposed for the seismic analysis of SRC structures. Firstly, the experimental studies on SRC frame columns

were conducted to study the hysteresis behavior under low cycle repeated loading. Based on the test results, the hysteresis behavior was analyzed, and the skeleton curve and hysteresis loops were further simplified. The simplified trilinear skeleton curve model was obtained through the corresponding feature points determined by regression analysis. Secondly, the nonlinear combination seismic damage index, which was developed by the lots test results and can well reflect the effect of the loading path and the number of loading cycles of SRC frame columns, was used to establish the cyclic degradation index. The strength and stiffness degradation rule of the SRC frame columns was analyzed further by considering the effect of accumulated damage caused by an earthquake. Finally, a hysteretic model of SRC frame columns was established and the specific hysteresis rules given. The proposed hysteretic model was validated by comparing the experimental and the numerical results on SRC frame columns.

## 2. Description of Test Program

In this paper, eight SRC frame column specimens were designed and constructed based on the current Chinese standards [53,54]. All of the specimens have the same cross-sectional shape and dimensions, however they have different compression ratios $n$ (=0.2, 0.4, 0.6), stirrups ratios $\rho_v$ (=0.8%, 1.1%, 1.4%), and steel rations $\rho_s$ (=4.6%, 5.7%, 6.8%). The configurations of the cross sections are shown in Figure 1. The details of all specimens are summarized in Table 1.

**Table 1.** Design parameters of specimens.

| Specimen | Cross Section (mm) | Aspect Ratio | Compression Ratios | Steel Shape | Stirrups Type |
|---|---|---|---|---|---|
| SRC-1 | 150 × 210 | 3.0 | 0.4 | I 14 | Φ6@110 |
| SRC-2 | 150 × 210 | 3.0 | 0.4 | I 14 | Φ6@110 |
| SRC-3 | 150 × 210 | 3.0 | 0.2 | I 14 | Φ6@110 |
| SRC-4 | 150 × 210 | 3.0 | 0.6 | I 14 | Φ6@110 |
| SRC-5 | 150 × 210 | 3.0 | 0.4 | I 10 | Φ6@110 |
| SRC-6 | 150 × 210 | 3.0 | 0.4 | I 12 | Φ6@110 |
| SRC-7 | 150 × 210 | 3.0 | 0.4 | I 14 | Φ6@80 |
| SRC-8 | 150 × 210 | 3.0 | 0.4 | I 14 | Φ8@120 |

All the steel shape was made of Q235 steel. The concrete strength was C80. The measured concrete cube average compression strength $f_{cu}$, axial compression strength $f_c$, and elasticity modulus of concrete $E_c$ are 83.9MPa, 75.49MPa and 42.042MPa, respectively. The measured mechanical properties of steel shape, longitudinal rebars, and stirrups are shown in Table 2.

**Table 2.** Mechanical properties of the steel and rebars.

| Properties | Steel Shape | | Stirrups | | Longitudinal Rebars Φ 10 |
|---|---|---|---|---|---|
| | Flange | Wed | Φ 6 | Φ 8 | |
| Yield strength $f_y$ (MPa) | 319.7 | 312.4 | 397.5 | 354.5 | 386.3 |
| Tensile strength $f_u$ (MPa) | 491.5 | 502.5 | 438.0 | 457.3 | 495.7 |
| Elasticity modulus $E_s$ (MPa) | $2.07 \times 10^5$ | | $2.07 \times 10^5$ | | $2.06 \times 10^5$ |

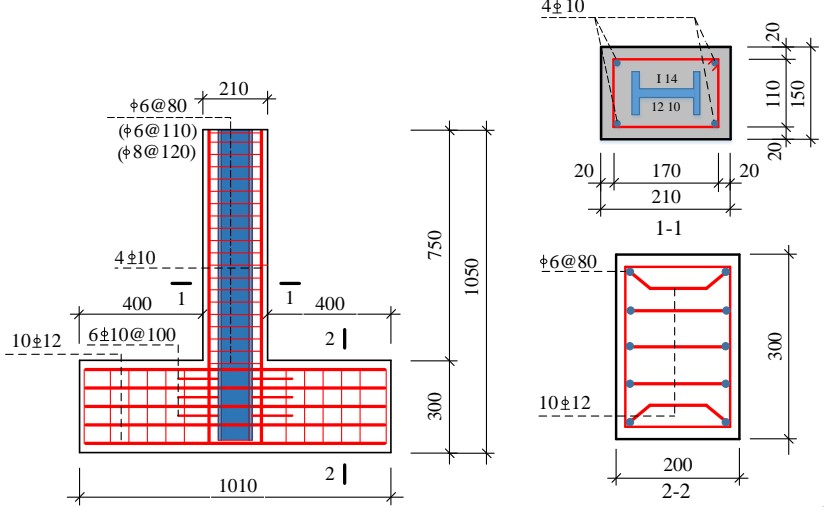

**Figure 1.** Cross-sectional dimensions and distributed steels of specimens (unit: mm).

A schematic and a photo of the test setup are shown in Figures 2 and 3, respectively. In order to prevent sliding of the specimen during the test, the specimen was fixed on strong ground by using two support steel beams and two fixed steel beams, which have a certain rigidity as a constraint of the frame column. Each specimen was tested under a constant axial gravity load during the experiment before applying a lateral force $P$ to simulate seismic loading. The lateral force $P$ was applied by an electro-hydraulic servo actuator placed at the top of the column. Linear variable displacement transducers (LVDTs) were adopted to monitor the horizontal displacement of the columns.

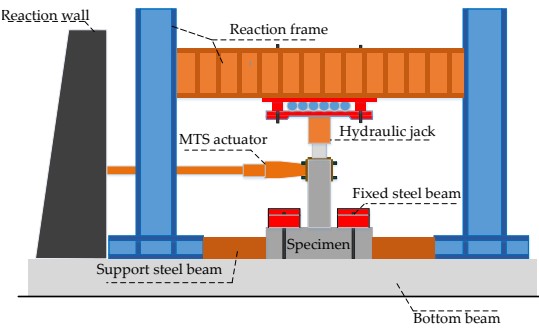

**Figure 2.** Test setup.

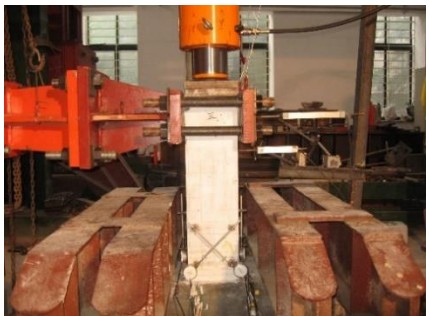

**Figure 3.** Photo of testing apparatus.

The monotonic loading was applied on the specimen SRC-1, and the remaining specimens were subjected to one cycle of loading with displacement ductility $\mu$ equal to 0.2, 0.4, 0.6, and 0.8, and followed by three successive cycles of loading with the displacement ductility $\mu$ equivalent to 1.0, 2.0, 3.0, . . . . until the horizontal load dropped to 80% of the peak load of the specimens or the load could

not be continued because damage was apparent and the loading was stopped [55]. The specific cyclic loading history is shown in Figure 4.

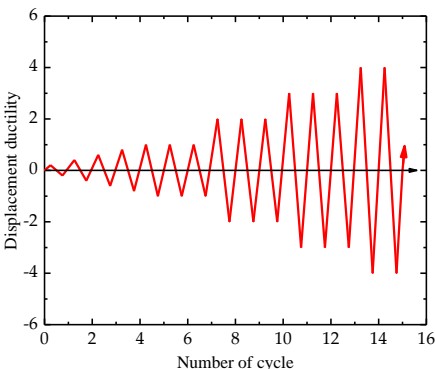

**Figure 4.** Loading history of specimens.

## 3. Analysis of Hysteretic Behavior

The loading-displacement ($P-\Delta$) hysteresis curves of the members under cyclic loading are the basis of the hysteretic performance for the structural members. Based on the test results, the loading–displacement hysteretic curves and skeleton curves for the SRC frame columns, which can be obtained by connecting the peak points of the hysteretic loops, are shown in Figure 5. It can be seen that:

1.  All of the test members have a similar failure process under repeat cyclic loading that experiences three failure stages, namely, the elastic stage, the inelastic stage with cracks, and the failure stage. At the initial stage of loading, several horizontal cracks appear first at the root of the column. The stress and strain of the rebars are small, and the action of the steel has not yet occurred. The specimen is mainly in the elastic stage. At this time, the hysteresis behavior of the SRC frame columns is similar to the reinforced concrete frame columns. Therefore, the hysteresis loops have different degrees of pinching phenomenon. With the increase of the displacement amplitude and the number of cycles, the cracks continually expand and develop. Meanwhile, the concrete cover appears severely cracked with partial spalling. At this point, because of the effect of the steel and core concrete, the pinching phenomenon of the hysteresis loops is gradually improved. When the displacement ductility $\mu$ reaches 3, some of the horizontal cracks develop into shear diagonal cracks, but the development speed of the diagonal cracks is relatively slow. The pinching phenomenon of the hysteresis loops disappears basically. At this time, the hysteresis loop becomes a plump fusiform shape. With continuous increase in load, the longitudinal rebars, steel flange, and most of the steel web yield at the crack section. Meanwhile, due to the action of the repeated load, the horizontal crack gradually runs through. The whole process is in the inelastic stage with cracks. Finally, with the increase of load further, the specimen comes into the failure stage. The concrete cover of the compression zone at the root of the column is a large spalling area. The stirrups and longitudinal rebars are exposed, and some of the longitudinal rebars buck. However, since the core concrete is constrained by a steel flange frame, the ability of compression deformation of the SRC frame column is obviously enhanced. Meanwhile, the strength and stiffness degradation of the members are slow, due to the constraint of the transverse stirrups on the concrete, there is a lateral bracing on the outside of the steel, which can prevent the global and local buckling of the steel. The whole process of the hysteresis cycle shows that there is a good ductility and an ability of energy consumption of the SRC frame column.
2.  Compared to the reinforced concrete members, the initial stiffness of the skeleton curve of the SRC frame column is larger due to the steel, and the declining part of the skeleton curve is relatively flat. At the same time, the peak load of the positive loading skeleton curve is slightly higher than the one of the reverse loading skeleton curve, since the specimen has a certain residual

deformation after positive loading. Under reverse loading, it is necessary to offset the influence of residual deformation in the members. In addition, positive loading generates a certain damage to the members, which reduces the bearing capacity of the members under repeated loading.

3.  With the increase of the axial compression ratio, the initial stiffness and peak load of the specimen gradually increase. However, after reaching the peak load, the larger the axial compression ratio, the faster the specimen strength and stiffness degrade. The energy dissipation capacity of the specimens significantly reduces, and the ductility becomes poor. With the increase of steel ratios, the peak load of the specimen gradually increases. After reaching the peak load, with the increase of the steel ratios, the strength degradation of members is relatively slow, and the ability of the ultimate deformation is strong. The stirrups ratio has little effect on the peak load by comparison of different stirrups ratios but the strength degradation of the specimen is relatively flatter with the increase of the stirrups ratios and the energy dissipation capacity and the ductility of members are enhanced.

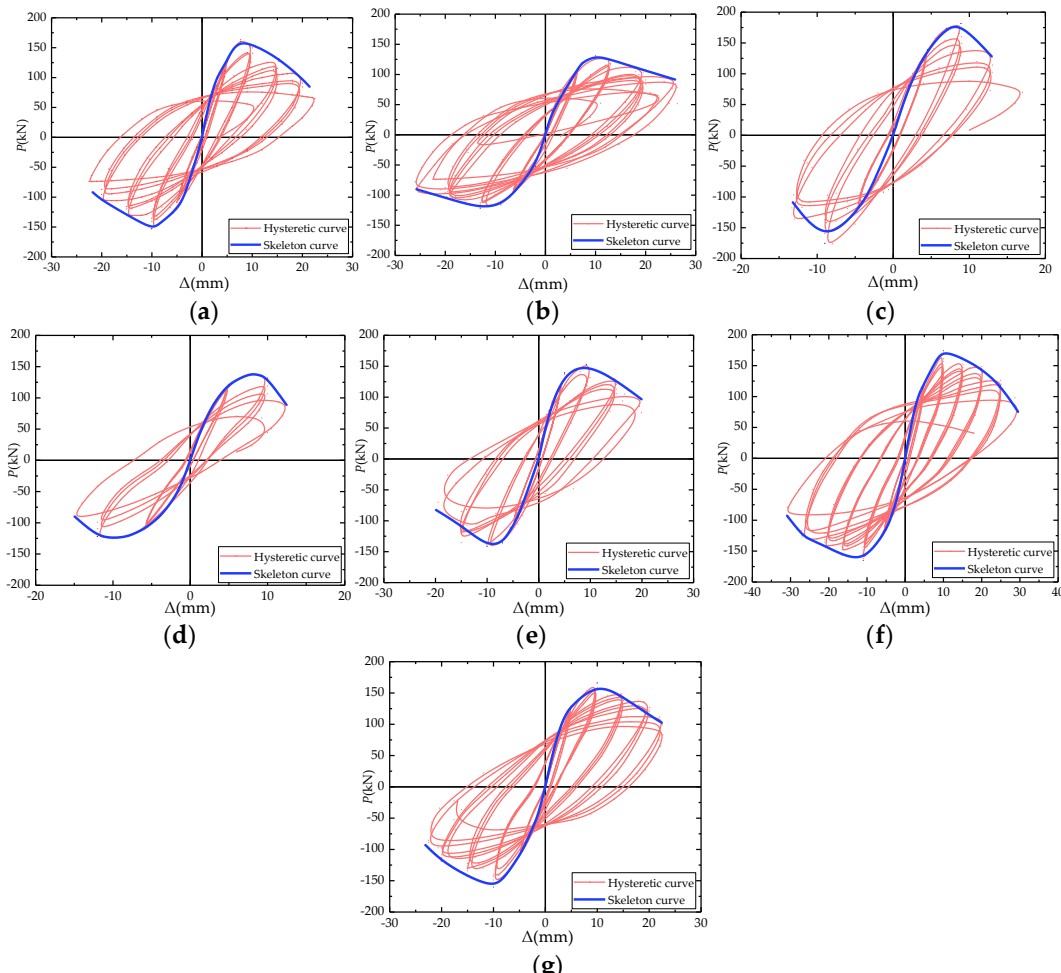

**Figure 5.** Hysteresis curves and skeleton curve of specimens. (**a**) SRC-2; (**b**) SRC-3; (**c**) SRC-4; (**d**) SRC-5; (**e**) SRC-6; (**f**) SRC-7; (**g**) SRC-8.

## 4. Determination of Skeleton Curve

### 4.1. Simplified Skeleton Curve

At present, most skeleton curves are obtained by connecting the peak points of the member hysteresis curves. In fact, the members would produce a certain degree of damage under the process of cyclic loading. With the increase of the number of cycles and displacement amplitude, the damage

accumulation of the members continuously increases. Therefore, it is difficult to determine the skeleton curve by the existing method and truly reflect the degradation of the mechanical properties of the members under an earthquake. A monotonic load–displacement curve can be used as a skeleton curve to better reflect the degradation of the structural members performance caused by cyclic loading and describe the hysteresis characteristics of structural members under an earthquake [56,57]. Therefore, in this paper, the load-displacement curve under monotonic loading of members was used as the skeleton curve as shown in Figure 6. Based on the results of this test, it can be seen that the SRC frame column slightly cracks before yielding. In addition, the lateral deformation of the member is smaller and the slope of the loading curve changes little. At the same time, the initial stiffness of the skeleton curve of the members is relatively large, there is no obvious inflection point when the specimen yields. The strength after yielding is still greatly improved. Therefore, at the initial stage of loading, the skeleton curves are basically coincident and the peak load is close. After the peak load, although the strength of the member decreases, the slope of the falling part is not large, which indicate that the strength degradation of the members is relatively slow. However, the damage accumulation of the members continually increases under repeated loading, which induces the degradation of the bearing capacity and stiffness, and the falling part of the skeleton curve becomes steeper. The ultimate deformation is reduced. Considering that the main purpose of the elastoplastic seismic response analysis is to study the mechanical behavior of the member after entering the inelastic stage, in order to simplify the calculation, in this paper, the skeleton curve before the yield of the SRC frame column can be simplified by a connection from the coordinate origin to the yield point. Therefore, the skeleton curve of the SRC frame column can be simplified to an ideal trilinear skeleton curve model that is shown in Figure 7. The segment OA is the elastic stage, and the segment AB that has a significant strength hardening section in the skeleton curve after the specimen yielded, is the elastic-plastic stage. In addition, the segment BC is the plastic descent segment.

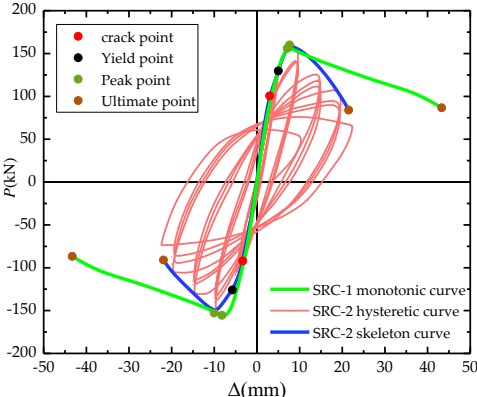

**Figure 6.** Typical monotonic and cyclic experimental response of the steel reinforced concrete (SRC) frame column.

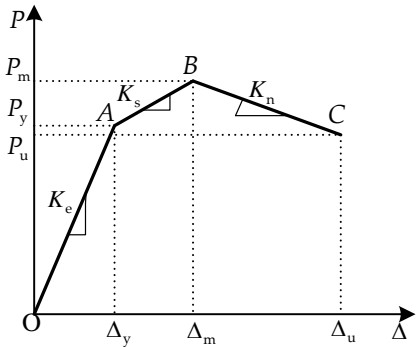

**Figure 7.** Simplified skeleton curve.

*4.2. Characteristic Points*

In order to obtain the proposed trilinear skeleton curve model of SRC frame column in this paper, the six quantities should be determined firstly: the yield load $P_y$, the yield displacement $\Delta_y$, the peak load $P_m$, the peak displacement $\Delta_m$, the ultimate displacement $\Delta_u$, and the ultimate load $P_u$.

The following assumptions are used to calculate the six quantities:

a) the section strain is subject to the plane section assumption.

b) the tensile effect of the concrete is ignored in the tension zone.

c) the steel and rebars have the same material properties under compression and tension, and are subject to ideal elastoplastic stress–strain relationship.

4.2.1. The Yield Load $P_y$ and Yield Displacement $\Delta_y$

(1) Large eccentric Compression Columns

For large eccentric compression, the yield point of the SRC columns corresponds to the steel yielding at the tension flange. According to the test results, when the longitudinal tension rebars yield, the steel tension flange is also close to yielding. The maximum strain of concrete $\varepsilon_0$ adopted in this paper is more than 0.002 that is larger than the yield strain of the longitudinal rebars and the steel flange. Therefore, it can be assumed that the concrete stress in the compression zone is a quadratic standard parabolic distribution, and the calculation diagram of the large eccentric compression SRC column and the stress–strain distribution of the section are shown in Figure 8.

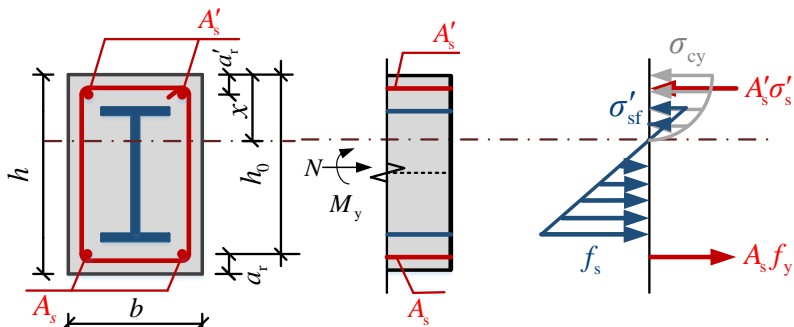

**Figure 8.** Calculation diagram of large eccentric compression at the yield state.

As shown in Figure 8, the section curvature when the root section of the frame column yields is defined as:

$$\varphi_y = \frac{\varepsilon_y}{h_0 - x}. \tag{1}$$

where $\varphi_y$ is the yield curvature of members, $\varepsilon_y$ is the yield strain of longitudinal rebars in the tensile zone, $h_0$ is the distance from the concrete compression edge to the center of the resultant force of the tension rebars, $x$ is the height of the concrete compression zone.

Key for calculating the yield curvature of the section is to determine the height of the concrete compression zone $x$, referring to Figure 8, the equations of internal force in the cross section are expressed as follow:

$$N + f_y A_s + f_s A_{sf} + \frac{1}{2} f_s t_w (h - x - a_s - t) = \sigma'_s A'_s + \frac{2}{3} \sigma_{cy} bx + \frac{1}{2} \sigma'_{sf} t_w (x - a'_s - t) + \sigma'_{sf} A'_{sf}, \tag{2}$$

$$\begin{aligned}
f_y A_s h_0 + f_s A_{sf}(h - a_s - \tfrac{t}{2}) + \tfrac{1}{6} f_s t_w (h - x - a_s - t)(x + 2h - 2a_s - 2t) + 0.5Nh \\
= M_y + \sigma'_s A'_s a'_r + \sigma'_{sf} A'_{sf}(a'_s + \tfrac{t}{2}) + \tfrac{1}{6} \sigma'_{sf} t_w (x - a'_s - t)(2x + a'_s + t) + \tfrac{1}{4} \sigma_{cy} bx^2
\end{aligned} \tag{3}$$

where,

$$\sigma'_s = \frac{x - a'_r}{h_0 - x} f_y \tag{4}$$

$$\sigma'_{\text{sf}} = \frac{x - a'_{\text{s}}}{h - x - a_{\text{s}}} f_{\text{s}},\tag{5}$$

$$\sigma_{\text{cy}} = \frac{x}{h_0 - x} \cdot \frac{k\varepsilon_{\text{y}}}{\varepsilon_0} f_{\text{c}},\tag{6}$$

$$k = \alpha(1 + \lambda_{\text{v}}).\tag{7}$$

where $N$ is the axial expression force, $f_{\text{s}}$ is the are the tensile strength of the steel, $f_{\text{y}}$ is the tensile strength of the longitudinal rebar, $A_{\text{s}}$ and $A'_{\text{s}}$ are the area of longitudinal tensile and compression rebar, respectively, $\sigma'_{\text{sf}}$ is the compressive stress of the steel, $\sigma'_{\text{s}}$ is the compressive stress of the longitudinal rebars, $\sigma_{\text{cy}}$ is the compressive stress of concrete edge when longitudinal bars yield under tension, $A'_{\text{sf}}$ and $A_{\text{sf}}$ are the cross sectional area of top flange and bottom flange of steel, respectively, $t_{\text{w}}$ is the thickness of the steel web, $t$ is the thickness of the steel flange, $b$ is the section width of the column, $h$ is the section height of the column, $a_{\text{s}}$ is the distance from the steel top flange to the column edge, $a'_{\text{s}}$ is the distance from the steel bottom flange to the column edge, $a_{\text{r}}$ and $a'_{\text{r}}$ are the distance from the center of gravity of the longitudinal rebar to the edge of the tension zone and from the center of gravity of the longitudinal rebar to the edge of the compression zone, respectively, $f_{\text{c}}$ is the concrete compression strength, $M_{\text{y}}$ is the section yield bending moment, $k$ is the strength enhancement factor considering the restraint effect of stirrups and steel on concrete, $\alpha$ is the consider the constraint effect of steel on core concrete, according to reference, $\alpha$ is equal to 1.1 in this research [53], $\lambda_{\text{v}}$ is the stirrups characteristic value.

The height of the concrete compression zone $x$ and the section yield bending moment $M_{\text{y}}$ can be calculated by Equation (2) and Equation (5), respectively.

(2) Small eccentric compression columns

For small eccentric compression, according to the test results, when the member yields, the steel compression flange has yielded or nearly yielded, the concrete stress at the edge of the compression zone also reaches the compressive strength of concrete, and the steel tension flange has not yet yielded. So, the calculation diagrams of the small eccentric compression SRC column and the stress– strain distribution of the section are shown in Figure 9.

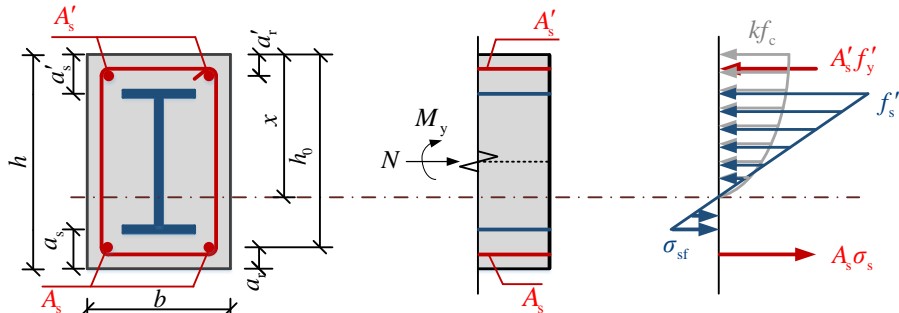

**Figure 9.** Calculation diagram of the small eccentric compression at yield state.

It can be seen from Figure 9 that the curvature of the section can be calculated as follows when the root section of the frame column yields,

$$\varphi_{\text{y}} = \frac{\varepsilon_0}{x}\tag{8}$$

According to Figure 9, the equations of the internal force in the cross section are given as follows:

$$N + \sigma_{\text{s}}A_{\text{s}} + \sigma_{\text{sf}}A_{\text{sf}} + \frac{1}{2}\sigma_{\text{sf}}t_{\text{w}}(h - a_{\text{s}} - t - x) = f'_{\text{y}}A'_{\text{s}} + f'_{\text{s}}A'_{\text{sf}} + \frac{2}{3}kf_{\text{c}}bx + \frac{1}{2}f'_{\text{s}}t_{\text{w}}(x - a'_{\text{s}} - t)\tag{9}$$

$$\sigma_s A_s h_0 + \sigma_{sf} A_{sf}\left(h - a_s - \frac{t}{2}\right) + \frac{1}{6}\sigma_{sf}t_w\left(h - a_s - \frac{t}{2} - x\right)(x + 2h - 2a_s - t) + 0.5Nh$$
$$= M_y + f'_y A'_s a'_r + f'_s A'_{sf}\left(a'_s + \frac{t}{2}\right) + \frac{1}{4}kf_c bx^2 + \frac{1}{6}f'_s t_w\left(x - a'_s - \frac{t}{2}\right)(x + 2a_s + t)\ , \tag{10}$$

where,

$$\sigma_{sf} = \frac{h - x - a_s}{x - a'_s}f_s, \tag{11}$$

$$\sigma_s = \frac{h_0 - x}{x - a'_r}f_y. \tag{12}$$

where $a'_s$ is the distance from the resultant force of longitudinal compression rebars to the edge of the compression zone, $f'_s$ and $f'_y$ are the compression strength of steel and longitudinal rebars, respectively, $\sigma_{sf}$ and $\sigma_s$ are the tensile stress of steel and longitudinal rebars, respectively, the rest of the symbols are the same as before.

The height of the concrete compression zone $x$ and the section yield bending moment $M_y$ can be calculated by Equation (9) and Equation (10), respectively.

Based on the practical stress condition of the frame column, the frame column can be simplified as a cantilever which bears axial pressure and horizontal force, as shown in Figure 10. Meanwhile, we can assume that the section curvature of the SRC column is linearly distributed along the column height. When the root section of the frame column hass yielded, the horizontal yield displacement of the top of the column can be expressed as:

$$\Delta_y = \frac{1}{3}\varphi_y H^2. \tag{13}$$

where $H$ is the distance from the loading point to the column root.

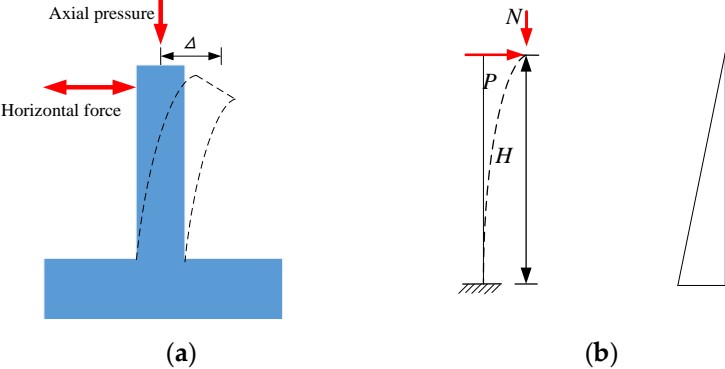

(a)                    (b)

**Figure 10.** Load and curvature distribution of the frame columns. (**a**) Force diagram; (**b**) distribution of curvature.

According to the force balance condition, when the root section of the frame column has yielded, the horizontal load of the top of the column is expressed as:

$$P_y = \frac{M_y - N\Delta_y}{H}. \tag{14}$$

### 4.2.2. Peak load $P_m$ and peak displacement $\Delta_m$

When the section reaches the maximum flexural capacity, the peak load of the frame column can be calculated by:

$$P_m = \frac{M_{max} - N\Delta_m}{H} \tag{15}$$

where $\Delta_m$ is the horizontal displacement corresponding to the peak load of the frame column, $M_{max}$ is the maximum flexural capacity at the root section of the frame column.

According to the simplified skeleton curve obtained above, the peak displacement $\Delta_m$ can be expressed as:

$$\Delta_m = \frac{P_m - P_y}{K_s} + \Delta_y. \tag{16}$$

where $K_s$ is the hardening stiffness, $K_s = \alpha K_e$, $K_e$ is the elastic stiffness, $\alpha_s$ is the hardening coefficient which has the relationship with the axial compression ratio $n$, $0.2 \leq n \leq 0.4$, $\alpha_s = 0.3$, $0.4 < n \leq 0.6$, $\alpha_s = 0.4$.

### 4.2.3. Ultimate Load $P_u$ and Ultimate Displacement $\Delta_u$

The ultimate load can be 80% of the peak load when the root of the frame column is failure, namely, $P_u = 0.8P_m$. The ultimate displacement can be expressed as follows:

$$\Delta_u = \Delta_m - \frac{P_m - P_u}{K_n} = \Delta_m - \frac{0.2P_m}{K_n}. \tag{17}$$

where $K_n$ is the softening stiffness, $K_n = \alpha K_e$, $\alpha_n$ is the softening coefficient which has the relationship with the axial compression ratio $n$, based on the statistical analysis of test data, $0.2 \leq n \leq 0.4$, $\alpha_n = -0.07$, $0.4 < n \leq 0.6$, $\alpha_n = -0.1$.

## 5. Damage-Based Hysteretic Analysis

### 5.1. Simplification of the Hysteresis Loop

An intact hysteresis loop consists of a loading curve and an unloading curve. According to the results of this test, after yielding of the members, the slope of the loading curve decreases with increasing displacement, indicating that the stiffness of the members has degraded during each repeated loading process. Like the loading curve, the slope of the unloading curve also decreases with the increase of the number of cycles, and the unloading stiffness of the members continuously degrades. After completing unloading, the member has some residual deformation that accumulates continuously with the increase of the number of cycles.

In order to better reflect the good seismic performance of the SRC members and enhance the simulation accuracy of the hysteresis curve in the later stage of loading, according to the results of this test and the characteristics of the hysteresis curves, the hysteresis loop is simplified as follow,

1. When the horizontal load does not reach the yield load, the hysteresis loop is simplified into three parts, namely, the elastic section OA (CD), the strengthening section AB (DE), and the unloading section BC (EF) in Figure 10.
2. When the horizontal load reaches the yield load, the hysteresis loop is simplified into four parts, namely, the elastic section HI (LM), the strengthening section IG (MN), the softening section GK (NO), and the unloading section KL (OP) in Figure 11.

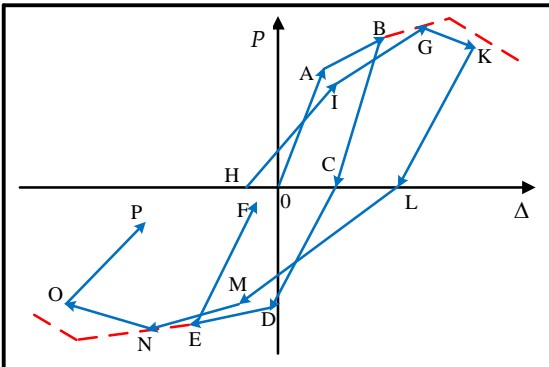

**Figure 11.** Simplified hysteretic loops.

### 5.2. Proposed Damage Index

To obtain the seismic damage index for SRC members, based on the test results of the SRC frame columns under low-cycle loads, the nonlinear combination seismic damage index, which can reflect the degradation of strength and stiffness under an earthquake, was developed by considering the effect of the loading path and the number of loading cycles [58]. This damage index is used in this paper to analyze the hysteretic behavior of the SRC column, and is expressed as:

$$D = (1 - \gamma) \sum_{j=1}^{N_1} \left( \frac{\Delta_{\mathrm{max},j} - \Delta_{\mathrm{y}}}{\overline{\Delta}_{\mathrm{u},i} - \Delta_{\mathrm{y}}} \right)^c + \gamma \sum_{i=1}^{N_{\mathrm{h}}} \left( \frac{E_i}{\overline{E}_{\mathrm{u},i}} \right)^c \tag{18}$$

where,

$$\overline{\Delta}_{\mathrm{u},i} = A + Be^{-\alpha}, \tag{19}$$

$$\overline{E}_{\mathrm{u},i} = A + Be^{-\alpha}, \tag{20}$$

$$c = 5.69 + 0.87 \ln \rho_{\mathrm{v}} + 0.056\lambda + 10.46\rho_{\mathrm{s}} - 2.1n. \tag{21}$$

where $D$ is the damage index, $\Delta_{\mathrm{max},j}$ is the largest inelastic deformation corresponding to the $j$-th half-cycle, $N_1$ is the number of half-cycles of $\Delta_{\mathrm{max},j}$ generated for the first time, $E_i$ is the hysteretic energy dissipation of the $i$-th half-cycle, $N_{\mathrm{h}}$ is the number of half-cycles, $\gamma$ is a composite parameter and is equal to 0.15 in this paper, $c$ is the experimental parameter, $\overline{E}_{\mathrm{u},i}$ and $\overline{\Delta}_{\mathrm{u},i}$ are the normalized ultimate energy dissipation capacity and ultimate deformation under constant axial load and monotonic lateral load after the $i$-th half-cycle, respectively, $\alpha$ is the normalized cycle cumulative energy dissipation under the $i$-th half-cycle, $\alpha = \sum_{i=1}^{N_{\mathrm{h}}} E_i / E_{\mathrm{u},0}$, $E_{\mathrm{u},0}$ is the ultimate energy dissipation capacity under monotonic loading directly, for ultimate energy consumption, $A = 0.46$, $B = 0.54$, and for ultimate deformation, $A = 0.76$, $B = 0.24$, $\lambda$ is the shear span ratio and equals 3.0 in this paper.

### 5.3. Cyclic Strength and Stiffness Degradation in the Hysteretic Model

#### 5.3.1. Damage-Based Cyclic Degradation Index

The cycle degradation index $\beta_i$ can be used to describe the degradation of the performance of the structures and members under an earthquake. The previous research results have shown that the loading path has a significant influence on the energy dissipation capacity of the members, therefore, the cyclic degradation index that was established by hysteretic energy dissipation cannot satisfactorily consider the influence of the loading path. Meanwhile, it is difficult to give the unified expression of component energy dissipation capacity under an earthquake. Therefore, in this paper, the cycle degradation index, which was based on the damage index obtained above, was proposed to better consider the effect of the loading path and cycle number on the seismic performance of the frame column, and it is expressed as:

$$\beta_i = [\Delta D_i / (1 - D_{i-1})]^\varphi \tag{22}$$

where $\Delta D_i$ is the increment of the component damage value for the $i$-th loading cycle; $D_{i-1}$ is the cumulative damage value of the member before the $i$-th loading cycle; and $\varphi$ is the correlation coefficient, based on the test results, and is equal to 1.2 in this research. Based on the degree of damage of the member under cyclic loading, the cyclic degradation index can describe the degradation of member performance. Meanwhile, the impact of the loading path on the cyclic degradation index is also considered. The value of the cyclic degradation index $\beta_i$ is between 0 and 1. When the value is closer to 1, it indicates that the degradation of the member performance is more serious. If $\beta_i < 0$ or $\beta_i > 1$, it means that the damage value increment of the member exceeds the residual damage value of the member under a certain cyclic loading, which indicates that the structural member is invalid. Therefore, the structure members failure criterion can be expressed as:

$$\Delta D_i > 1 - D_{i-1}. \tag{23}$$

In the following, the degradation rule of the member performance is described by using the damage-based cyclic degradation index $\beta_i$.

**Remark:** The damage index (Equation (18)), the cyclic degradation index (Equation (22)), and the associate failure criterion (Equation (23)) are hysteresis based. Both are obtained based on the hysteresis curves through destructive tests. However, currently there is a lack of studies to link the sensor-based damage index with the hysteresis-based damage index through destructive tests. It would be interesting to conduct tests of SRC specimens integrated with embedded piezoceramic smart aggregates and to correlate these two indices. If these two indices correlate well, the health status of an in-service structure can be monitored in real-time through embedded sensors based on the sensor enabled damage index. In addition, the failure criterion (Equation (23)) can help to establish the corresponding criterion for the sensor-based damage index.

### 5.3.2. Degradation Analysis of Strength and Stiffness

(1) Degradation of yield load

With the increase of the number of load cycles, the yield load of the member decreases continuously after each reverse loading and reloading, and the degradation rule of the yield load is:

$$P_{yi}^{\pm} = (1 - \beta_i) P_{y(i-1)}^{\pm}. \tag{24}$$

where $P_{yi}^{\pm}$ is the yield load of the member after the *i*-th cycle loading, and $P_{y(i-1)}^{\pm}$ is the yield load of the member before the *i*-th cycle loading. The superscript "+" means positive loading, "−" means reverse loading, and the same is shown below.

(2) Degradation of hardening stiffness

With the increase of the number of load cycles, the hardening stiffness of the members is degenerated, and the degeneration rule is:

$$K_{si}^{\pm} = (1 - \beta_i) K_{s(i-1)}^{\pm}. \tag{25}$$

where $K_{si}^{\pm}$ is the hardening stiffness of the member after the *i*-th cycle loading, and $K_{s(i-1)}^{\pm}$ is the hardening stiffness of the member before the *i*-th cycle loading.

(3) Degradation of softening stiffness

With the increase of the number of load cycles and the displacement, the softening stiffness of the member gradually degrades and approaches to the origin, the degradation rule of softening stiffness can be determined by the following equation:

$$K_{ni}^{\pm} = (1 - \beta_i) K_{n(i-1)}^{\pm} \tag{26}$$

where $K_{ni}^{\pm}$ is the softening stiffness of the member after the *i*-th cycle loading and $K_{n(i-1)}^{\pm}$ is the softening stiffness of the member before the *i*-th cycle loading.

Figure 12 shows the degradation rule of member yield load, hardening stiffness, and softening stiffness. The member is loaded from point 0 to point 2 along the loading path 0–1–2. The initial loading stiffness is $K_e$, the hardening stiffness is $K_{s0}^{+}$, the softening stiffness is $K_n^{+}$. Then along point 2 unloaded to point 3, the unloading path is 2–3. A half cycle is completed from point 0 to point 3 according to Equation (18) to calculate a damage value. Then the degradation index $\beta_i$ is calculated through Equation (22), and the respectively corresponding strength and stiffness values are calculated by Equation (24), Equation (25), and Equation (26). In the reverse loading, the yield load of the members decreases from $P_y^{-}$ to $P_{y1}^{-}$, the slope of the hardening stiffness decreases from $K_{s0}$ to $K_{s1}^{-}$, and the slope of the negative stiffness decreases from $K_n^{-}$ to $K_{n1}^{-}$. From point 6 unloads to point 7, the

damage value is calculated by Equation (18) again. Next the half-cycle damage value increment $\Delta D_i$ is calculated, and then the degradation index $\beta_i$ is recalculated by Equation (22). When the member is reloaded along the load path 7–8–9, the yield load of the structure reduces from $P_y^+$ to $P_{y1}^+$, the hardening stiffness decreases from $K_{s0}$ to $K_{s1}^+$, and the softening stiffness decreases from $K_n^+$ to $K_{n1}^+$.

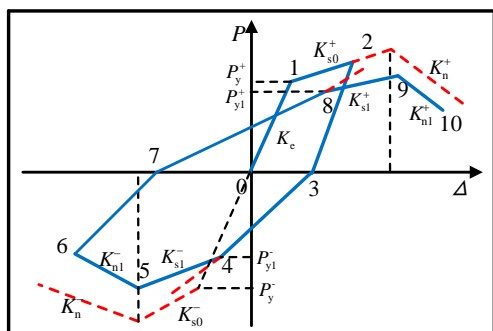

**Figure 12.** Degradation of yield load, hardening stiffness, and softening stiffness of members.

(4) Degradation of unloading stiffness

The existing research results show that the unloading stiffness of the member does not significantly degrade during the elastic phase and the elastoplastic stage, which is almost the same as the initial stiffness $K_e$. When the horizontal load reaches the peak load and the structure is in the plastic force state, the unloading stiffness of the members degenerates and can be described as follows:

$$K_{ui} = (1 - \beta_i)K_{u(i-1)} \tag{27}$$

where $K_{ui}$ is the unloading stiffness of the member after the $i$-th cycle loading, and $K_{u(i-1)}$ is the unloading stiffness of the component before the $i$-th cycle loading.

The unloading stiffness degradation of the member in the plastic phase is shown in Figure 12. The loading starts from point 0 to point 2 along the loading path 0–1–2. At this time, according to Equation (18), we can calculate a damage value. Then, according to Equation (22), we can calculate the degradation index $\beta_i$. The unloading stiffness is calculated through Equation (27), and the unloading stiffness decreases from the initial stiffness $K_e$ to $K_{u1}$ at this time. When reverse loading reaches point 5, the damage value is calculated by Equation (18) again. At the same time, the damage value increment $\Delta D_i$ is calculated from the unloading point 2 to reloading point 5. Then, the deterioration index $\beta_i$ is recalculated by Equation (22), and the unloading stiffness is calculated by Equation (27). At this time, the unloading rigidity of the member decreases from $K_{u1}$ to $K_{u2}$.

(5) Degradation of reloading stiffness

Based on the assumptions of the previous hysteresis loops, the reloading stiffness is related to the unloading stiffness of the previous cycle. The degradation rule of reloading stiffness can be expressed by the following equation:

$$K_{ri} = (1 - \beta_i)K_{ui}. \tag{28}$$

where $K_{ri}$ is the reloading stiffness of the member after the $i$-th cycle loading.

As shown in Figure 13, the loading starts from point 0 to the reverse loading point 3 along the loading path 0-1-2-3. At this time according to Equation (18) to calculate a damage value, and then according to Equation (22) to calculate the degradation index $\beta_i$. During the reverse loading, according to the Equation (28), the reloading stiffness of the member is changed from $K_e$ to $K_{r1}$, and then the reverse loading starts from point 3 to the reverse unloading point 6 along the loading path 3-4-5-6. At this time according to Equation (18) to calculate a damage value. At the same time, the damage increment $\Delta D_i$ of the half cycle is calculated, and then the degradation index $\beta_i$ is recalculated by Equation (22). During reloading process, the reloading stiffness of the member is changed from $K_{r1}$ to $K_{r2}$.

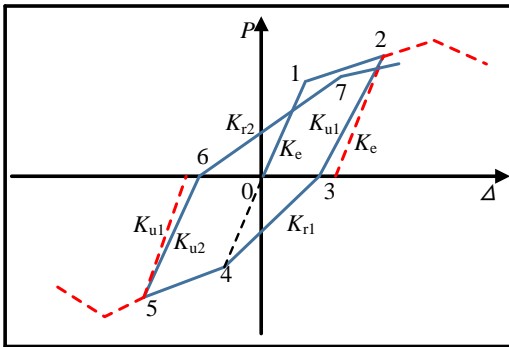

**Figure 13.** Degradation of unloading stiffness and reloading stiffness of members.

### 5.4. Hysteresis Rules

Based on the above description of the performance degradation of the member, the hysteresis rules of the SRC frame column can be summarized as follows:

1.  Before the component has not yielded, the loading and unloading are performed along the elastic segment of the member skeleton curve.

2.  After the load born by the member reaches the yield load, the loading path is performed along the skeleton curve of the member. During the unloading process, the corresponding damage value is calculated by Equation (18) at the unloading point. At the same time, the corresponding degradation index $\beta_i$ is calculated by Equation (22), and the unloading stiffness is calculated by Equation (27).

3.  The reverse loading and reloading path: after a half-cycle is completed, the damage value of the member is recalculated, and the half-cycle damage value increment $\Delta D_i$ is calculated. Then, the degradation index $\beta_i$ is calculated according to Equation (22). The reverse loading stiffness, the yield load, the hardening stiffness, and the softening stiffness of the member hysteresis loop are calculated by Equation (28), (24), (25), and (26) respectively before the loading starts. The continued loading is performed along the softening stiffness of the members. The stiffness is calculated as before, and the path at the time of reloading is the same as described above.

## 6. Validation of Hysteretic Models

To verify the effectiveness of the hysteretic model established in this paper, the hysteresis curves calculated by this method were compared to the hysteresis curves obtained by the experiment, and the results are shown in Figure 14, where the data of the specimen SRC-6 and SRC-11 are used from the reference [59]. It can be seen that the results of calculation and test are in good agreement with each other, which indicates that the hysteretic model of the SRC frame columns proposed in this paper can better describe the hysteresis characteristics of the SRC structural members under an earthquake.

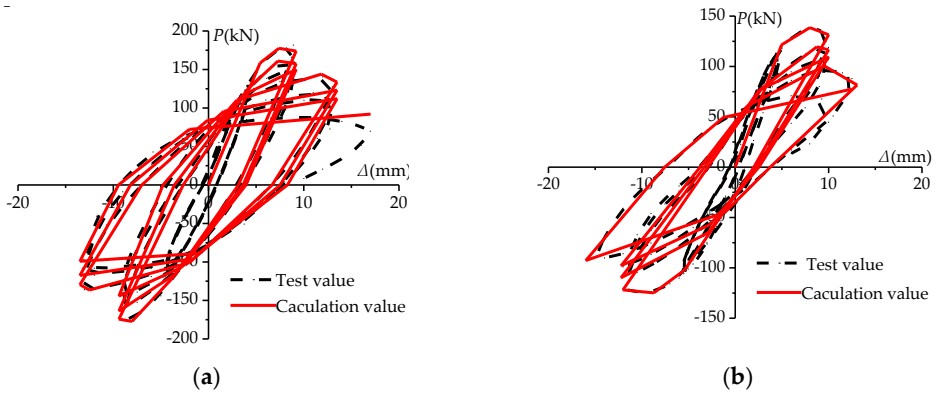

(**a**)                           (**b**)

**Figure 14.** *Cont*.

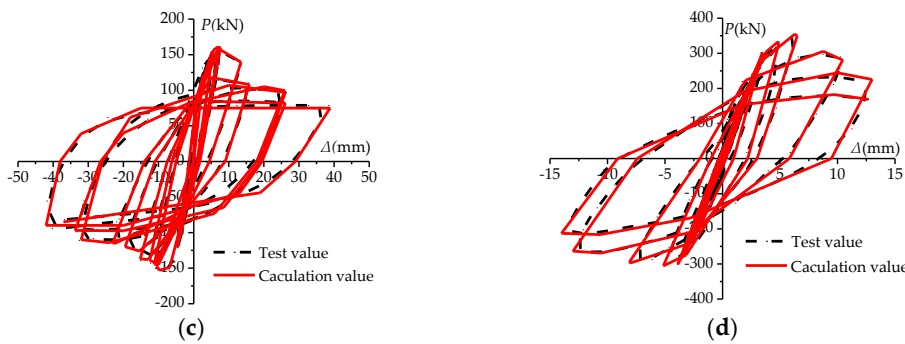

**Figure 14.** Comparison of testing and calculating hysteretic curves. (**a**) Specimen SRC-4; (**b**) Specimen SRC-5; (**c**) Specimen SRC-6; (**d**) Specimen SRC-11.

## 7. Conclusions and Future Work

In this paper, eight SRC frame column specimens were tested under combined axial compression and lateral cyclic load. Base on the test results, a hysteresis model was developed to simulate the hysteresis behavior of the SRC frame columns, and conclusions were obtained as follows:

1. The failure process of the SRC column specimen experiences three stages under the cycle load, namely, the elastic stage, the inelastic with cracks stage, and the failure stage. In the elastic stage, the hysteresis behavior of the SRC frame columns is similar to the reinforced concrete frame columns. In the inelastic stage, due to the mutual constraint between steel and concrete, the bearing capacity of steel and concrete is improved. With the increase of the displacement amplitude and the number of cycles, the hysteresis loops become a plump fusiform. Especially, after the peak load, the specimens show a good anti-seismic performance.

2. With the increase of the axial compression ratio and steel ratio, the seismic performance of the frame column is enhanced. The stirrups ratio has the little effect on the seismic performance of the frame column before the peak load. However, after the peak load, the energy dissipation capacity and the ductility of the members is enhanced with the increase of the stirrups ratios.

3. Based on the comparison of the test results of the monotonic loading and cycle repeat loading, a trilinear skeleton curve model of the SRC columns can be established by simplifying the load-displacement curve under monotonic loading. According to the equilibrium of the internal force, a set of calculation equations for determining the simplified skeleton curves of the SRC columns are proposed.

4. The hysteresis loop is simplified by combining with the experimental results. Based on the seismic damage index of the SRC frame columns, the cyclic degradation index is established, which can reflect the performance degradation of the members very well. Finally, the multi-line hysteresis model of the SRC frame column is established by establishing the rule of strength and stiffness degradation of the members and the hysteresis rule. It was shown that the computed hysteresis curves are in good agreement with the experimental results, which verifies the validity of the hysteresis model. The proposed model is able to predict the cyclic response of SRC columns with sufficient accuracy.

Future work will involve testing SRC specimens with embedded piezoceramic smart aggregates. During the tests, we will obtain both the sensor-based damage index and the hysteresis-based damage index and study the correlation between the two indices. If the two indices correlate well, we can use the sensor-based damage index to provide the real-time data of structural health monitoring for in-service structures with integrated sensors. Meanwhile, the proposed method in this paper has a low computational efficiency, which is not convenient for real seismic analysis of structures. Therefore, based on the cumulation damage of the material and the results proposed in this paper, we will conduct finite element analysis (FEA) of the SRC structure members in future investigations. Additionally, the stress state of concrete confined by stirrups and steel is more complicated for SRC

members. Therefore, to achieve the refined finite element simulation of the SRC structures, based on the cyclic constitutive model for concrete confined by transverse reinforcements [60,61], the cyclic constitutive model of confined concrete established by considering the effect of cumulation damage will be studied in future research.

**Author Contributions:** B.W., G.H., and Y.S. designed and performed the experiments; B.W. and S.Z. analyzed the data; B.W. and G.H. wrote the paper.

**Funding:** This research was funded by the National Science and Technology Support Program: 2013BAJ08B03; National Natural Science Foundation of China: 51678475; the Natural Science Foundation of Shaanxi Province: 2018JQ5158; Scientific Research Program was Funded by Shaanxi Provincial Education Department: 18JK0382.

**Acknowledgments:** The authors appreciate the support of the National Science and Technology Support Program (2013BAJ08B03), the National Science Foundation of China (51678475), the Natural Science Foundation of Shaanxi Province (2018JQ5158) and the Scientific Research Program Funded by Shaanxi Provincial Education Department (18JK0382).

**Conflicts of Interest:** The authors declare no conflict of interest.

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
