# Peer review of "Hysteretic Behavior of Steel Reinforced Concrete Columns Based on Damage Analysis"

_applsci, doi:10.3390/app9040687_

Round 1

Reviewer 1 Report

This is a good paper and it can be accepted after following revisions:

1. Authors need to consider following related references in the introduction and discussion section:

Cyclic constitutive model for high-strength concrete confined by ultra-high-strength and normal-strength transverse reinforcements. Australian Journal of Structural Engineering, 12(2), 59-72.

Stress–strain model for concrete under cyclic loading. Magazine of Concrete Research, 36, 330-347.

2. Authors need to add a parametric analysis section. In this parametric analysis section following parameters should be considered: the column height to the beam span, axial compression ratio, the steel ratio of column, the steel ratio of beam and the longitudinal reinforcement ratio of beam.

3. FEM analysis should be used to compare with the developed method.

Author Response

1. Authors need to consider following related references in the introduction and discussion section:

Cyclic constitutive model for high-strength concrete confined by ultra-high-strength and normal-strength transverse reinforcements. Australian Journal of Structural Engineering, 12(2), 59-72.

Stress–strain model for concrete under cyclic loading. Magazine of Concrete Research, 36, 330-347.

Response : The authors greatly appreciate the reviewer’s comment. The suggested references have been quoted and considered in the manuscript.

2. Authors need to add a parametric analysis section. In this parametric analysis section following parameters should be considered: the column height to the beam span, axial compression ratio, the steel ratio of column, the steel ratio of beam and the longitudinal reinforcement ratio of beam.

Response : The comment is considered; however, only the influence of different design parameters on hysteretic behavior was considered in this paper. The influence of design parameters on seismic performance of SRC structures has been reported in the author's previous articles as follows,

1. Zheng Shansuo, Wang Bin, Li Lei and Hou Piji. Study on Seismic damage of SRHSC frame columns. SCIENCE CHINA Technological Sciences: series E, 2011, 54 (11), 2886-2895.

2. Zheng Shansuo, Wang Bin, Hou Piji, Guo Xianfa, Yu Fei and Zhang Hongren. Experimental study of the damage of SRHSHPC frame columns under low cycle reversed loading. China civil engineering journal, 2011, 44(9), 1-10.

Therefore, in this paper, the influence of design parameters on seismic performance of SRC structures is not mentioned again.

3. FEM analysis should be used to compare with the developed method.

Response : This is a valuable comment. However, considering the manuscript is over 20 pages, the authors will conduct the FEM analysis in future investigations. Additionally, the authors have added a few discussions to clarify this issue in the section 7.

Reviewer 2 Report

This paper presented a new PH hysteretic model by considering the accumulated damage. the Hysteresis behavior was analyzed, and the skeleton curve and hysteresis loops were simplified based on the test results. The nonlinear combination seismic damage index was developed by the test results. It can well reflect the effect of the loading path and the number of loading cycle of SRC frame columns. It was also used to establish the cyclic degradation index. This paper established the hysteretic model of SRC frame columns. Some interesting results were found and the relevant theory was also given to analyze the results. The outcome of this study provided an important reference to the elastoplastic time-history analysis of structures. This paper is well written, but some issues need to be addressed. I recommend accepting the article if the following minor points are addressed.

1.    The ordinates of figure 5 should be set to be consistent to show differences of different specimens.

2.    All the steel shape was made of Q235 steel, but the yield strength of the steel shape in table 2 are 319.7 and 312.4, why?

3.    Is the word “steel rations” in the description of test program the spelling mistakes of “steel ratios”?

4.    What is the basis for loading mechanism?

5.    The stirrups are preferably indicated in red in figure 1(1-1).

Author Response

1. The ordinates of figure 5 should be set to be consistent to show differences of different specimens.

Response : The authors appreciate the reviewer’s comment. The ordinates of Figure 5 have been re-organized in the revised manuscript to show differences of different specimens.

2. All the steel shape was made of Q235 steel, but the yield strength of the steel shape in table 2 are 319.7 and 312.4, why?

Response : The authors greatly appreciate the reviewer’s comment. Q235 indicates the standard value of yield strength. In this paper, the yield strength of the steel shape in table 2 was obtained by the test directly. According to references [53, 54], the yield strength obtained by test should be no less than 125% of the standard value. Thus, the results are believable.

3. Is the word “steel rations” in the description of test program the spelling mistakes of “steel ratios”?

Response : The comment is agreed. Sorry for typo. It should be “steel ratios”.

4. What is the basis for loading mechanism?

Response : The comment is appreciated. According to the Reference [54], the yield displacement, yield load, ultimate load and ultimate displacement of the member are preliminarily determined, and then the ductility coefficient is obtained. Finally, the loading step, which has been added in the manuscript, is determined according to the Reference [55].

5. The stirrups are preferably indicated in red in figure 1(1-1).

Response : The authors appreciate the reviewer’s comment. The color of stirrups in Figure 1 (1-1) has been changed to red.

Round 2

Reviewer 1 Report

Paper can be accepted.